# Towards a Functional Performance Validation Standard for Industrial Low-Back Exoskeletons: State of the Art Review

**DOI:** 10.3390/s21030808

**Published:** 2021-01-26

**Authors:** Mattia Pesenti, Alberto Antonietti, Marta Gandolla, Alessandra Pedrocchi

**Affiliations:** 1Department of Electronics, Information and Bioengineering, Politecnico di Milano, 20133 Milan, Italy; alberto.antonietti@polimi.it (A.A.); alessandra.pedrocchi@polimi.it (A.P.); 2Department of Mechanical Engineering, Politecnico di Milano, 20156 Milan, Italy; marta.gandolla@polimi.it

**Keywords:** low back, industrial exoskeleton, assistive device, fatigue relief, worker support, low-back pain

## Abstract

While the research interest for exoskeletons has been rising in the last decades, missing standards for their rigorous evaluation are potentially limiting their adoption in the industrial field. In this context, exoskeletons for worker support have the aim to reduce the physical effort required by humans, with dramatic social and economic impact. Indeed, exoskeletons can reduce the occurrence and the entity of work-related musculoskeletal disorders that often cause absence from work, resulting in an eventual productivity loss. This very urgent and multifaceted issue is starting to be acknowledged by researchers. This article provides a systematic review of the state of the art for functional performance evaluation of low-back exoskeletons for industrial workers. We report the state-of-the-art evaluation criteria and metrics used for such a purpose, highlighting the lack of a standard for this practice. Very few studies carried out a rigorous evaluation of the assistance provided by the device. To address also this topic, the article ends with a proposed framework for the functional validation of low-back exoskeletons for the industry, with the aim to pave the way for the definition of rigorous industrial standards.

## 1. Introduction

Assistive devices have been spreading for decades now in several and heterogeneous fields. Indeed, this growth regards both the industry and the healthcare sector, with particular attention to the growing field of exoskeletons. Exoskeletons were firstly developed for human augmentation, helping the wearer to lift heavy objects [1]. This often resulted in bulky, heavy, and power-hungry devices characterized by the trade-off between force augmentation and freedom of motion. The technological development of the latest years allowed researchers to improve the user experience and the reliability of such assistive devices. Hence, exoskeletons also started to be developed for rehabilitation aims. Wearable robots, indeed, can support the relearning of motor control strategies in the brain [2,3,4]. This is achieved by means of movement repetitions characterized by the same trajectory and velocity.

The Lokomat [5] was the first case of a rehabilitation exoskeleton. Developed in the early 2000s, it was worn on the lower limbs and meant to assist the gait of stroke patients on a treadmill. Afterward, several other devices have been designed and commercialized for this purpose, both for treadmill and overground gait, such as the ReWalk [6]. In the meantime, exoskeletons for military use started to be developed, mainly to substantially improve the load-carrying capabilities of humans. A review of the historical background and recent developments on military, civil, and medical exoskeletons can be found in [7,8].

Nowadays, the research interest for exoskeletons is spread into several fields. In particular, it has been recently shifting from the medical/rehabilitation field to the industrial sector. This is due to several reasons. On the one hand, the development of rehabilitation exoskeletons may have reached a *plateau*, since reliable and efficient solutions are available for such applications, at least for what concerns the mechanical design. On the other hand, Industry 4.0 (i.e., the fourth industrial revolution) is leading towards the concept of smart factories. The adoption of automation in the industry has been growing for the last twenty years, intending to increase productivity while reducing the physical effort required to human workers. Nevertheless, robots cannot still achieve the high dexterity of humans, which makes them irreplaceable for many tasks. Indeed, in the European Union, the industrial sector is still among the ones with the highest work intensity, according to the sixth European Working Conditions Survey [9]. Forty-three percent of workers in EU28 are subject to tiring or painful positions one-quarter of the time or more, while 32% of them are exposed to carrying or moving heavy loads. As a consequence, the most widely reported health problem by workers is backache (or low-back pain), often causing absence from work. Moreover, work-related musculoskeletal disorders (WRMD) dramatically affect the quality of life of a large portion of the world’s population. Although there are signs of modest betterment of the working conditions, this report suggests that there is still a large margin for further improvements.

Exoskeletons for industrial use have the aim to help the workers with most of the tiring tasks. In particular, low-back exoskeletons might reduce the physical effort during manual handling of heavy loads and prolonged unsafe body postures. Alternatively, industrial workers may benefit either from upper-limb exoskeletons—such as MATE (Comau; Torino, Italy) or shoulderX (SuitX, US Bionics; Emeryville, CA, USA)—or from lower-limb exoskeletons—such as legX (SuitX, US Bionics; Emeryville, CA, USA). Recently, some comprehensive reviews on exoskeletons for the industry have been published [10,11], focusing on industrial applications of exoskeletons and their mechanical features and characteristics. Here, in  light of the analysis of the working conditions for the industry found above, we focus on low-back exoskeletons for industrial use only. In particular, we believe that low-back exoskeletons are one of the most promising solutions to effectively reduce the impact of WRMDs, in particular low-back pain due to intense physical activities in the workplace, as discussed above.

While the design of exoskeletons for industrial use is gaining research interest, the rigorous evaluation of these devices, in terms of assistance to the end users, has not been properly tackled yet. Indeed, the lack of a validation standard for exoskeletons is among the main factors limiting their adoption in the industrial field. Missing standards makes it difficult to compare the performance of commercially available devices. On top of that, validation standards for industrial exoskeletons must obey different regulations across the world. This very urgent issue is starting to be acknowledged and discussed by the researchers of the field. In [12], de Looze et al. analyzed 26 industrial exoskeletons (mostly for the upper body), also focusing on the effects they have on reducing the physical workload for their end users.

The aim of this paper is two-fold. On the one hand, we provide a systematic review of the state of the art of the functional performance evaluation methodology for exoskeletons designed to assist the low back. In particular, we focus on the validation of such class of devices in terms of *functional performance*, reporting the evaluation criteria and the metrics used for such validation, as well as the obtained results. On the other hand, starting from the results of such analysis, we suggest a set of criteria to evaluate the performance of low-back exoskeletons for industrial use in terms of assistance provided to the user.

The remaining of this paper is structured as follows. Section 2 illustrates the methods used to carry out this review. Section 3 presents the industrial low-back exoskeletons here analyzed, while Section 4 reports the results of the analysis on performance and evaluation metrics. In Section 5, we discuss some of the most important outcomes of this literature review, and we also propose a validation framework (Section 5.1) to assess and quantify the assistance of low-back industrial exoskeletons. Finally, Section 6 concludes the paper.

## 2. Materials and Methods

### 2.1. Search Methods for Data Collection

We defined the most relevant keywords for the electronic search of the studies to be analyzed. As shown in Table 1, they are divided into three categories, referring to: (i) the actuation of the device; (ii) the device type; and (iii) the assisted body part. While we are interested in low-back exoskeletons, it is quite common to refer to such devices with different terms, such as *trunk-bending wearable robot* or *hip exoskeleton*. In this way, we tried to make the search as comprehensive as possible. Exclusion criteria (see Section 2.2) allowed us to restrict the results to our interest after the query. The selected keywords were combined using Boolean operators (AND/OR). Specifically, OR combinations were applied among the elements of each column, while the AND operator was applied among the columns.

### 2.2. Query Results and Exclusion Criteria

The following online databases were used as primary source of data:PubMed (www.ncbi.nlm.nih.gov/pubmed);Web of Science (www.webofknowledge.com).

We restricted our search to documents published between 2009 and June 2020, applying a filter directly in each search engine. The combination of the keywords resulted in $N_{0}=905$ documents. First, we removed 105 duplicates by checking the Digital Object Identifier (DOI) of each item. Then, we applied the exclusion criteria listed in Table 2. These account for duplicates, false positives, i.e. articles that are not about exoskeletons, or studies that involved the usage of exoskeletons for different body parts. We also excluded articles that presented only results obtained in a simulated environment, or with impaired/disabled subjects. Finally, we discarded literature not available in English and other review articles.

A first screening was performed by reading the titles to remove obviously irrelevant studies. A second screening was done reading also the abstracts. The application of exclusion criteria resulted in the selection of 90 articles for full-text reading. Then, we further restricted the field to low-back exoskeletons and performed an in-depth analysis of the studies focusing on their assistive performance and how it was measured and validated. Specifically, we analyzed N=23 articles for this review. The data collection and selection process is reported in the PRISMA flowchart shown in Figure 1.

### 2.3. Data Analysis

The selected 23 articles were analyzed focusing on the following two aspects.
(i)Technical features of each assistive device. We report their most important characteristics, such as actuation type, number and type of their degrees of freedom (DoF), and assisted tasks.(ii)Characteristics of the validation strategy. We analyzed study type and contribution and the evaluation criteria and metrics used for functional validation. With the term *functional validation*, here we intend the evaluation of the effectiveness of the device in assisting the end user, thus reducing their physical workload. Again, we do not foresee nor search for any carryover effect induced by the device (i.e., rehabilitation effect) or any raw physical power increase (i.e., human augmentation effect).

## 3. Low-Back Exoskeletons

In this section, we report the analysis of the state of the art regarding assistive devices for the low back. In this definition, we include both exoskeletons and powered orthoses. In the reviewed literature, most of the assistive devices designed to support the low back of industrial workers are exoskeletons. Indeed, out of the 23 studies analyzed, 22 used exoskeletons, and only 1 used an orthosis. We found 12 different devices, of which 11 are exoskeletons, and 1 is the aforementioned orthosis. Of these devices, six are commercially available and six are academic prototypes. Many of the back-support exoskeletons that are commercially available (e.g., the ALDAK (Cyber Human Systems, Traña-Matiena, Spain), the AWN-03 (ActiveLink, Panasonic; Nara, Japan), the H-WEX (Hyundai Motor Company, Seoul, South Korea), and many more) were not found in this state-of-the-art search focused on the evaluation of the provided assistance. This was pointed out also in a recent review article [8] by Young and Ferris. The authors highlighted how issues related to intellectual property (IP) and confidentiality limit or even discourage scientific publications regarding their designs. The IP lock-up problem creates gaps in the industry–academia knowledge transfer and in the scientific literature of many research fields [13].

An overview of the main characteristics of the assistive devices is reported in Table 3. In the first column, we report either the name of the device or a short description of it, along with a reference to the presentation article of that device, when available, or its website. Commercially-available devices are marked with an asterisk (*) next to their name. For each device, we also show the number of studies in which it was found in the analyzed dataset (absolute frequency, *f*). Then, we list the tasks for which the device is designed to help the end users with. For the actuation type, while active and passive, respectively, stand for powered (robotic) or unpowered devices, the term *hybrid* (or semi-active) refers to the integration of both active and passive actuation technologies (e.g., as in a Series-Elastic Actuator scheme [14]). We also list the total number of degrees of freedom (DoF), specifying both the assisted ones (A-DoF) (either passive or active, depending on the device) and the free (or non-actuated) degrees of freedom (N-DoF). Additional free DoFs are a fundamental feature in terms of user experience and wearing comfort, given that the assistance should not hinder ordinary movements. Finally, for powered devices, we report the embedded sensors and the adopted control strategy. The former is intended as human-monitoring sensors, and the latter includes only the high-level control loop aimed at improving the human-machine interface. We do not include here low-level control systems and the sensors solely functional to those.

### 3.1. Actuation and Mechanical Design

Low-back exoskeletons are assistive devices designed to support the wearer with tasks that stress particularly the lumbo-sacral joint and the low region of the back. This assistance is required because of the intense compression forces that act on the spinal cord while performing actions typical of the industrial workplace, such as load lifting or prolonged static forward bending.

In the 23 reviewed articles, the SPEXOR and the Laevo (Laevo B.V., Delft, The Netherlands) are the most frequently reported exoskeletons. Both were found six times, while the Hybrid Assistive Limb for Lumbar Support (HAL, Cyberdyne; Tsukuba, Japan) was counted two times. All the other devices were found only once in the reviewed articles. Passive assistance for low-back exoskeletons is overall found in 50% of the analyzed devices (6 out of 12), while the rest are either active (4) or hybrid (2). Active devices may be preferred when bursts of high power are required, while passive devices may be more adequate when relatively low power delivery is required throughout the whole working day, as in the case of static forward bending. Hybrid exoskeletons may represent a trade-off capable of guaranteeing both high power delivery to actively-actuated degrees of freedom (compared to passive exoskeletons) and high power efficiency (compared to active exoskeletons) thanks to their passive components. On the other hand, the majority of the analyzed studies (15 out of 23) considered passive exoskeletons, outnumbering both active (6) and hybrid (2) designs. Rather interestingly, these figures suggest there is an increasing adoption of passive exoskeletons for industrial use, which have the advantages of being lightweight, cheaper, and easier to maintain compared to active devices.

As shown in Table 3, exoskeletons aimed to assist the low back generally have only one degree of freedom (1 DoF) per side, which corresponds to the flexion-extension at the level of the hip (denoted as α-Hip hereafter). In the case of bilateral assistive devices, the count of the degrees of freedom hereby reported is intended as DoFs per side. The action of the device at this joint is to assist with the flexion-extension of the trunk. Regarding the low back, the only two exceptions are the SPEXOR [15] and the ‘Lifting Assist Device’ (LAD) [16]. The former is a passive exoskeleton that integrates a second DoF at the L5-S1 joint to assist the flexion-extension of the spine (denoted as α-L5-S1). This additional assisted DoF aims to reduce the compression force that acts on the low back. The latter implements a similar design but with hybrid actuation. On the other hand, the ‘Lower-limb exoskeleton’ extends to the lower limbs, with one actuated DoF at the knee and two non-actuated DoFs (N-DoF) at the ankle. Such mechanical design is aimed to support the wearer with load carrying. Few of the analyzed exoskeletons feature additional N-DoFs. Indeed, the APO and the LAD are the only devices that feature at least one additional rotation on the low back. Differently from the case of the ‘Lower-limb exoskeleton’ [17], these N-DoFs have the aim to improve the wearability and the freedom of motion.

Low-back exoskeletons assist the wearer by means of a torque at the level of the hip, with the output power ranging from 25 to 160 Nm (see Table 3). This quantity depends on several design factors, among which actuation type, desired power-to-weight ratio, and maximum percentage compensation of the load are worth mentioning.

Most of the assistive devices here analyzed discharge the load to the ground through the body of the wearer. The only exception is the ‘Lower-limb exoskeleton’ that is designed to discharge the load directly to the ground at the level of the feet of the wearer. Wearable exoskeletons need to be designed with care, since they add a biomechanical load to the human body, independently of the provided assistance. The weight of the device and the reaction forces at the contact points between the device itself and the wearer must be carefully considered. Moreover, the muscular activity of the whole body should be studied during the development of the device or its validation, as non-target muscles (e.g., leg or abdominal muscles) could manifest an increased activity due to the usage of the device (see Section 4.2 for details).

### 3.2. Sensors and Control

Active and hybrid exoskeletons may be designed to automatically modulate the assistive torque they output as a function of one or more signals measured either on the wearer or on the device itself. In the case of industrial low-back exoskeletons, we identified two main categories in terms of control strategy to assist the wearer. One is based on the muscular activity level, and the other is based on task kinematics or dynamics. In the former case, electromyography (EMG) electrodes can be placed on the skin in correspondence with target muscles known to be active or under stress during the execution of the tasks to be assisted. EMG-based control strategies are found in the HAL, the Robo-Mate, and the hybrid APO. On the other hand, control strategies based on task kinematics typically exploit either inertial measurement units (IMU) or other sensors (inclinometers, potentiometers, etc.) to measure absolute and relative angles and then retrieve the posture of the human body. This information may be used to provide an assistive torque proportional to the inclination angle (i.e., torque ∝α-Hip).

It is interesting to discuss here the case of the Robo-Mate, in which a hybrid EMG+IMU control scheme is proposed. In this case, the two control strategies provide a torque that is proportional to each control signal, combined in a weighted sum. Potentially, this solution could allow the best assistance to the wearer, since it enables to adapt the output torque both to the kinematics and the muscular effort, with the drawback of higher costs and encumbrance.

The control system of the HAL combines EMG and inertial data in a different way. The Cybernetic Voluntary Control (CVC) provides EMG-based motion intention recognition: over-threshold EMG activity is used to provide coordinated support to the wearer. Simultaneously, the Cybernetic Autonomous Control (CAC) provides inclination-based gravity compensation, thanks to data acquired from embedded angular sensors and a triaxial accelerometer.

Regulating the torque of the exoskeleton in a proportional way to the inclination angle results in a much simpler control law with respect to EMG-based control. Indeed, in this control scheme, the extensor moment at the hip generated by the device is proportional to the angle of trunk flexion. This control scheme makes the robotic exoskeleton behave similarly to a passive one, usually assisted by means of springs. Thus, controlling the device with the EMG approach may result in a better and finer tuning of the assistance, as a function of the effort of the human wearer, thus improving the user experience. On the other hand, EMG-based control comes with some critical aspects to consider. These are the problem of electrode positioning, which affects signal quality and inter-trial variability, hence control robustness; the problem of the skin-electrode interface, which is sensitive to stress, environmental conditions, sweat, and other factors that have impact on the signal (and signal-to-noise ratio) and induce time-varying dynamics; and, finally, muscle fatigue—which manifests as a reduction of the muscular activation signal—should be well identified and counteracted in the control system in order to maintain an adequate performance of the device.

## 4. Assistance Evaluation

The evaluation of exoskeletons is a critical problem. It regards both the design phase of the exoskeleton itself and its spread with real end users. For the former, the proper metrics and criteria should be considered to tune the design and reach the desired requirements. For the latter, the possibility to robustly measure and validate the performance of one exoskeleton is fundamental both to assess the assistance it provides to the wearer and to compare it with alternatives available on the market. In the literature hereby analyzed, many different types of evaluation criteria are used to assess the performance of the tested exoskeletons. In the following, we focus on the validation approach of the N=23 analyzed articles.

### 4.1. Study Type and Contribution

In Table 4, we assign to each article an identifier and we report its bibliographic reference. When coming to the evaluation, it is important to consider the experimental setup and testing conditions of each study in order to better understand its aim and reliability. Thus, we report the testing scenario, i.e., whether the tasks are carried out in a laboratory (LAB), a living lab (LL), or in a real-world setting (RW), and the enrolled test subjects, who can either be end users (workers, with or without low-back pain) or healthy volunteers. Finally, we report the type of study and the contribution it provides, also indicating the exoskeleton used in that study. The type of study can be defined according to three categories. For *Proof of Concept*, we mean a demonstration of a new device with tests on subjects that differ from the end users of that device. A *Pilot* is instead a study in which the device is tested with subjects, either end users or healthy volunteers, repeating one or more tasks with and without wearing the exoskeleton in order to evaluate its functional performance, and measuring proper outcomes. Compared to a proof-of-concept study, the pilot has generally a larger number of both test subjects and metrics for functional evaluation, and follows more rigorous protocols for such evaluation. Finally, an *RCT* is a Randomized Control Trial type of study with end users. Similarly, we identified three categories for the contribution of each study. A *Feasibility analysis* is a study aimed to assess whether a device is suitable to assist the end users with a certain task. Instead, an *Effectiveness analysis* regards a qualitative or quantitative analysis of a device and its biomechanical evaluation, i.e., the evaluation of its *performance* in terms of assistance provided to the wearer. Finally, a *User-acceptance analysis* is a subjective evaluation of a device carried out with end users.

Most of the articles here analyzed discuss pilot studies that report the effectiveness analysis of a device. Specifically, 17 out of 23 are categorized as *pilot* for the type of study, while 14 are categorized as *effectiveness analysis* for the contribution of the study. It is worth mentioning here also the test–retest study (#6). The authors suggested a battery of tasks to evaluate low-back exoskeletons and reported results for a test–retest repetition using the SPEXOR. The same battery of tasks was used in other of the studies here analyzed, namely #5, #12, and #15.

These figures about the type of study are also reflected by choice of the experimental setup. Indeed, only 1 study out of the 23 analyzed was carried out in a real-world scenario, with 2 set up in living labs and the remaining 20 carried out in a laboratory. Similarly, there is a strong unbalance regarding the selected test subjects. In 16 of the analyzed studies, healthy test subjects were recruited to evaluate the exoskeleton under analysis. Tests with potential end users were carried out either with healthy workers (3/23), workers with low-back pain only (2/23), or with mixed groups (2/23). A detailed numerical report about the experimental setup, the type of study, and its contribution is given in Table 5.

The majority of these studies, namely 15 out of 23, were carried out with healthy subjects in a laboratory. These combinations of environment and test subjects often result in rather controlled experimental conditions. These are a potential source of bias for the results of such a study. As evident from the data reported here, even the few tests with potential end users were carried out in a laboratory setup or a living lab. Nevertheless, the validation of exoskeletons with end users usually requires a great effort for the researchers as well as for the enrolled subjects. On-field testing of exoskeletons is fundamental to assess the *real-world performance* of the device. Indeed, it may bring out some critical aspects that are generally not relevant in laboratory testing. For example, freedom-of-motion limitations due to the non-actuated DoFs (or lack thereof) of the device could be evident only in the RW setting. Another aspect worth mentioning is the long-term use of the device, considering the effects of both continuous usage in a work-shift and day-by-day repetitive usage. On the other hand, laboratory-based experiments are still fundamental for exoskeletons and their validation, as they could pave the way for further testing of such devices in more realistic settings.

### 4.2. Evaluation Criteria and Metrics

In this section, we focus on the evaluation criteria and the metrics used to validate the assistive performance of low-back exoskeletons for industrial use. As discussed throughout this manuscript, with the term *performance validation*, here we mean the assessment and the rigorous quantification of the assistance provided by the exoskeleton to the human user, measured according to the selected criteria. Objective metrics are fundamental to evaluate the functional performance of an exoskeleton, comparing the results obtained across repetitions of the same task executed with and without the device. On the other hand, subjective evaluation criteria are important as well for a device. Wearing comfort, pain reduction, ease of use, and many other criteria are part of the user experience of the device. Positive feedback from the end users of the device is fundamental for its success and spread in the industry.

To better report this analysis and compare the selected studies, we introduce here five domains in which we categorize those criteria.
Muscular: EMG-based metrics measured to evaluate a change of muscular activity due to the use of the exoskeleton.Force/torque: Computation of the compression force acting on the L5-S1 joint or the flexion-extension moment about that joint; joint net torque; mechanical joint work; and ground reaction force (GRF).Metabolic: Measurement of the metabolic cost or metabolic rate and derived quantities.Functional: Task-related metrics, such as measurements of kinematics, performance time, posture holding time, repetition count, walking distance (carrying a payload or not), and so on.Subjective: Perceived task difficulty (PTD), measures of system usability and acceptability, perceived effort, and pain measures.

In the reviewed literature, we found 25 evaluation criteria and metrics. The muscular and functional domains were the most common ones, as metrics of both these domains were found in 15 out of 23 studies. Metrics belonging to the subjective domain were found in 12 articles. Finally, the least common domains are the force/torque and the metabolic ones, as metrics belonging to these were found in seven and three studies, respectively. For each domain, in Table 6 we report a list with all the evaluation criteria and metrics found in the analyzed articles, along with their absolute frequency, i.e., the number of studies in which that metric was found. In absolute terms, the most frequently used metrics are the muscle activity (muscular domain), found in 15 out of 23 studies, and the measurement of kinematic variables (functional domain), found in 11 studies.

### 4.3. Muscular Domain

The measurement of muscle activity by means of EMG is the most common technique used to evaluate exoskeleton assistance, as shown in Table 6. Rather intuitively, the more assistance is provided by the exoskeleton and the more muscle activity reduction should be measured. This can be quantified, for example, as the relative (percentage) variation of EMG-based metrics computed while test subjects perform a task with and without the exoskeleton. The root mean square (RMS) computation is often applied to EMG data. Either the peak or the time-average value of the RMS data can be considered, depending on the task. Peak values are generally preferred for dynamic tasks, such as load lifting, while average values are generally used for static tasks, such as static forward bending. The processing of EMG measurements to obtain muscular activation signals is quite standardized. Band-pass filtering, rectification, and normalization with respect to the maximum voluntary contraction (MVC) value are common steps found in the vast majority of the literature about EMG signals, even in several other fields. RMS computation can be applied after the normalization of the signal.

As mentioned, muscle activity is the most frequently used metric. On the other hand, evaluating the assistive performance of an exoskeleton using only peak or average muscle activity values may be not exhaustive. Indeed, this approach may fail to capture the muscular effort exerted by the wearer throughout the whole task, as well as its overall intensity. These limitations could be overcome by measuring the muscular effort variation due to the exoskeleton with the time integral of the muscle activity (iEMG), found in two of the analyzed studies. The integrated EMG activity can be thought of as a measure of integrated force over time [43]. Thus, it may be more suitable to compare the overall muscular effort required to execute a certain task with and without the exoskeleton, better evaluating the assistance provided by the device.

In the case of low-back exoskeletons for industrial use, EMG electrodes can be placed in correspondence of the erector spinae (ES) complex. In Figure 2, we provide two illustrations that show the muscles of the human body that were found in the reviewed literature. As shown in the posterior-body view (Figure 2a), the group of muscles of the erector spinae run bilaterally in parallel to the spine.

Therefore, electrode positioning to measure the activity of this complex should be done carefully and well documented for reproducibility. In many of the reviewed articles, this is done according to the European recommendation for surface electromyography (SENIAM, www.seniam.org) [44]. On the other hand, many other studies do not report exhaustive details about electrode positioning, making it difficult to either replicate or compare such results.

In Table 7, we report all the muscles mentioned in the reviewed literature. The activity of these muscles was there used to evaluate the assistance provided by the exoskeleton. As shown, not only muscles of the low back are found, but also some of the lower limbs and abdominal ones. For every single muscle or muscle group, we report its absolute frequency. Muscles of the same complex are grouped together, as in the case of the erector spinae, for which we also show sub-groups with their relative sub-muscles. The iliocostalis lumborum (IL), for example, is part of the erector spinae iliocostalis (ESI), which in turn is part of the erector spinae complex.

As mentioned, while low-back muscles are the most frequent in the reviewed studies, other body parts are considered for muscle activity analysis as well. For example, muscles of the thigh, such as the biceps femoris, are often considered. The reason behind this choice is to investigate the potential stress that could be added to the muscles of the legs due to the added weight of the exoskeleton and the reaction forces, due to its functioning, that discharge to the ground through the human body (see Section 3.1). Similarly, the muscles of the anterior part of the body are also quite frequently considered. As shown in Figure 2b, these mostly include, again, muscles of the thigh as well as abdominal muscles, such as the quadriceps femoris or the rectus abdominis, respectively.

### 4.4. Functional Domain

In the functional domain, the most commonly used evaluation metrics are the measurement of kinematic variables and the performance time. The former is meant to measure eventual restrictions in the range of motion (RoM) of the human joints while wearing the exoskeleton, as well as alterations of the physiological trajectories while performing a certain task. The latter, on the other hand, is focused on task performance, as it measures any differences in the time of execution for a certain task. While an increased task execution time could result in a negative user experience and productivity reduction, its decrease could be unwanted as well. In fact, increasing the rate of task execution could increase the physical effort and stress for the workers aided by the exoskeleton. Ideally, exoskeletons for human assistance, differently from human-augmentation devices, should allow the wearer to keep a similar task-execution pace while the device reduces the required effort.

It is worth mentioning other metrics found in the functional domain, such as the repetition count and the load-carrying walking distance. Low-back exoskeletons are often optimized for static forward bending and/or load lifting and, more in general, to handle payloads with a weight of up to 20 kg. On the other hand, load-carrying tasks are often neglected either in the validation of the exoskeleton or even in the design phase. This can result in exoskeletons that hinder walking even without any additional payload. In #6, a battery of 12 tasks is proposed to evaluate the assistive performance of the SPEXOR. Specifically, the authors divided these tasks into two groups, namely tasks for which the exoskeleton is expected to aid the wearer and tasks for which the exoskeleton is expected to hinder the wearer. The latter group includes, for example, walking and climbing a ladder. This confirms that low-back exoskeleton can potentially hinder some tasks, such as load carrying, thus requiring a functional evaluation of this aspect.

### 4.5. Metabolic Domain

In the metabolic domain, the metabolic cost [J/kg/s] is the only metric found in the analyzed literature. On the other hand, this kind of metric is more frequently used in the case of lower-limb exoskeletons for gait support. Nevertheless, the evaluation of the metabolic cost has been found in a few analyzed studies (namely, 3 out of 23). Typically, this quantity is derived from direct measurements of breath using a gas analyzer. Variations of either oxygen consumption or CO${}_{2}$ production are typically exploited to compute the metabolic cost during task execution. The exoskeleton is expected to take over some mechanical work that would be generated by the muscles, thus reducing the metabolic cost required to the human wearer [28]. A reduction of the mechanical work produced by the low-back muscles could in turn result in a reduced compression force at the L5-S1 joint. Hence, a reduction of the metabolic cost measured during load lifting could suggest a reduction of the stress applied to the spinal cord. Therefore, the analysis of metabolic cost could also give insights on muscular activity and mechanical stress on the low back. On the other hand, other techniques to measure this quantity should be considered using some of the metrics found in the force/torque domain.

### 4.6. Force/Torque Domain

Metrics of the force/torque domain are biomechanical quantities that estimate the stress to which the musculoskeletal system is subject during task execution. The most important ones are the (peak) compression force and the (peak) flexion-extension moment at the lumbo-sacral (L5-S1) joint. Indeed, while lifting a 15 kg payload, the load on this joint can be up to 5000 N [15]. The consequence of such intense mechanical solicitations on the spinal cord is low-back pain. For this reason, measuring how much the flexion-extension moment or the compression force at the L5-S1 joint can be reduced is fundamental for industrial low-back exoskeletons. Indeed, these evaluation metrics can give insights on how much the device can prevent or reduce low-back pain in industrial workers.

L5-S1 flexion-extension moments and compression forces are typically computed based on the ground reaction force (GRF) and lower-body kinematics, using a bottom-up inverse dynamical model. This typically requires using a force plate for GRF measurement, thus potentially limiting the application of these metrics to laboratory-based setups. The state-of-the-art model for this computation, found in the large majority of the analyzed studies, is the one reported in [45]. On the one hand, the usage of a common method to compute such variables helps to compare the validation results among different studies. On the other hand, differences in experimental conditions, tasks, and study population still make it difficult to effectively compare different exoskeletons based on the available literature.

### 4.7. Subjective Domain

Metrics of the subjective domain aim to evaluate the user experience and the general impression of the wearer in terms of comfort/discomfort and provided assistance. Subjective questionnaires generally have the aim to understand whether the exoskeleton is capable of reducing perceived difficulty and effort during task execution. In the analyzed literature, the most frequently used metrics of the subjective domain are the perceived task difficulty (PTD), the rating of perceived exertion (RPE) measured according to the BORG-CR10 scale, and subjective measures of perceived discomfort, either local (i.e., associated to specific body parts, such as low back or chest) or general. Hence, subjective metrics are useful to measure the wearing comfort, which may influence task execution, and the perceived musculoskeletal effort, which may influence the perceived pain, stress, and fatigue. Although many different metrics of this kind exist, they are generally evaluated using a visual analog scale (VAS), provided to the subjects after the execution of a certain task with and without the exoskeleton. All the subjective metrics found in the reviewed articles are reported in Table 6 with their absolute frequency.

Subjective metrics are fundamental to evaluate how the wearer perceives the provided assistance. In #18, the authors evaluated the Laevo with subjective metrics only. The peculiarity of this study is that the evaluation is repeated over time. When measured after four weeks with respect to the initial evaluation, some subjective ratings of perceived discomfort and usability worsened. As shown in Table 8, the workers testing the exoskeleton found greater local perceived discomfort (LPD) at the chest and reported lower task performance, thus resulting in a 25% decrease of the intention to use the exoskeleton in the workplace.

### 4.8. Performance Analysis and Comparison

In Table 8, we list the results reported in the analyzed literature in terms of functional performance evaluation. For each study, we list the numerical identifier (as defined in Table 4), the device under evaluation, the tasks executed during its evaluation, and the number of subjects that participated in the study, also indicating their gender. Studies that used the same exoskeleton are grouped together to facilitate quantitative comparisons. Finally, we indicate the results reported in 22 out of the 23 analyzed articles (note that we do not include here the test–retest study (#6) as the authors did not report the results of single trials), reporting for each metric the corresponding domain, as defined above (see Section 4.2). This table allows analyzing the results reported in the reviewed literature and compare them.

In Figure 3, we provide a visual representation of the distribution of the evaluation criteria and metrics with respect to their domains across the analyzed studies. For each study, we specify the number of metrics used in the performance evaluation per each domain. This allows immediately perceiving the sparsity of such distribution in the analyzed literature, thus highlighting the lack of a validation standard.

In terms of muscle activity, all the analyzed studies reported results that are in agreement with each other. The activity of low-back muscles is reduced by the exoskeleton of reasonable amounts in all the studies, most of the time ranging from −10% to −40%. Several factors influence the exact amount of this reduction, among which the actuation of the exoskeleton (active/passive/hybrid), the executed task, and the test subjects are the most relevant. Similar considerations can be done for the compression force and the flexion-extension moment at the lumbo-sacral joint. Again, the entity of this reduction depends on the experimental setup.

Although found in only three studies, metabolic cost reductions are consistent, with an average reduction of around −17.5%. In particular, this average reduction of the metabolic cost is found for payload lifting. On the other hand, the metabolic cost of load carrying is found to increase because of the exoskeleton in #9. This increase was also dependent on the walking speed, which was reported to be slower with the exoskeleton (+12% metabolic cost) compared to the natural walking speed of the subjects (+17%).

The metrics of the functional domain are the ones that vary the most among the analyzed studies. The angle of trunk flexion (α-Hip), for example, is reported to be reduced when wearing the exoskeleton (up to −15.9%) in some studies (#4, #9, and #10), while it is unchanged (#2 and #4) or even increased (#14) in others. Rather interestingly, this also happens considering the same exoskeleton and similar tasks. For example, the Laevo is evaluated both in #9 (static forward bending) and #14 (static forward bending; static posture holding). In the former case, the authors report a significant decrease in the trunk-bending angle. Conversely, a 15.9% significant increase of the same variable is reported in the latter case. The vast majority of studies reports reduced values of α-Hip. Contradictory results in other studies may be due to differences in test population and executed tasks. Moreover, studies with a larger sample size should be considered to obtain more robust results.

Finally, it is interesting to notice that the results reported for the subjective domain metrics are often in agreement among different studies. Indeed, test subjects across different studies reported improvements for static forward bending and load lifting, expressed as lower PTD and discomfort in the low back. Similarly, they reported a worsening of the same metrics, i.e., an increase of task difficulty and discomfort, for other tasks, such as walking (either carrying a load or not) or squatting.

#### Performance Comparison

Considering the studies here analyzed, we can compare the validation of different exoskeletons by the same research groups, as well as the results reported by different studies for the same exoskeleton. This comparison is intended as a *semi-qualitative* analysis because of all the limitations discussed throughout this manuscript.

The first case reported here is a direct comparison between the Laevo and the BackX AC found in #7. This is the most reliable comparison, as both the experimental setup and the test population are the same. As reported in Table 8, the BackX AC obtained better results in terms of muscle activity reduction, with an average 35% reduction compared to the average -13% obtained with the Laevo during static forward bending. Similarly, the authors reported a worsening considering the performance time for both exoskeletons only in the case of female subjects. Conversely, male subjects enrolled in the study reported a significant 75% increase of the perceived musculoskeletal effort (BORG CR-10) using the BackX. This comparison is a clear example of the importance of user feedback for industrial exoskeletons. Even though the objective performance of the BackX are superior to the ones of the Laevo, the negative user feedback of the latter may limit its user experience and thus its adoption in the industry.

There is another type of performance comparison that can be considered in this review. This regards the performance evaluation of the same exoskeleton by different research groups. This analysis aims to understand whether different researchers consider similar evaluation metrics and, in that case, notice if the obtained results match or not.

The performance of the Laevo can be compared considering two of the studies (#3 and #7) here analyzed. Both the study population and the assisted tasks are different. In #3, a group of 11 male subjects as enrolled to perform load lifting. The Laevo is here evaluated with metrics of the muscular, force/torque and functional domains. In #7, on the other hand, a gender-balanced group of 18 subjects performed static forward bending with and without the exoskeleton. The authors chose metrics of the muscular, functional and subjective domain. In #3, an 8% reduction of the average muscle activity is reported considering five muscles (IL, LL for the low back, RA, EO, IO for the abdomen). In #7, the authors considered a similar set of muscles (TES, IL and RA, EO), but then computed two metrics to evaluate the assistance of the exoskeleton. Indeed, they reported a 22.5% reduction of the activity of the TEM (trunk erector muscles), computed as the average of TES and IL, as well as a 13% reduction of the TTM (total trunk muscle activity), computed as the average of all the acquired muscles. Although it is not possible to compare single muscles, the slightly higher average reduction reported in #7 may be due to the aforementioned differences in tasks and study population. The results reported for the metrics of other domains cannot be compared.

Another comparison of this kind can be done for the HAL, used in both #11 and #20 for load lifting. The two studies used metrics of different domains. In #11, the BORG-CR10 is used to evaluate the effect of the HAL on the perceived muscular effort, reporting no significant differences. Again, the subjective evaluation of the effort appears to be in contrast with the achieved reduction of muscle activity, measured as a significant decrease of both EMG and iEMG for both ES and QF (see Table 8). On the other hand, in #20, the authors reported a 25% decrease of the perceived lumbar fatigue when using the HAL. Moreover, when lifting at a fixed pace, subjects obtained a +44% and a + 45% increase for lifting time and number of lifts, respectively.

The last performance comparison is the one provided in #6. As mentioned, this paper provides a reliability study in which the authors set up a test–retest trial with the same exoskeleton (the SPEXOR). The results of this study were not considered in the previous analysis, as the authors reported only intra-session and inter-session reliability scores [46] for a battery of 12 tasks used to evaluate the exoskeleton. On the one hand, the intra-session reliability considers two repetitions of such tasks, comparing both functional and subjective metrics between these two repetitions. On the other hand, the inter-session one compares the scores obtained in the first test with a repetition done 7–10 days later. Intra-session reliability was found to be good or excellent for all tasks (which include static forward bending, lifting, carrying, kneeling, and sit to stand), except for load carrying and lifting, for which it was moderate. Inter-session reliability scores were contaminated by systematic effects, for which participants reported lower discomfort scores and lower task difficulty at the second test session. This caused lower inter-session reliability scores for some tasks, such as load lifting, with respect to the intra-session reliability. Hence, this study suggests that caution is needed when interpreting long-term differences in the outcome of individual tasks aided by exoskeletons.

## 5. Discussion

In this section, we point out and emphasize some crucial aspects that emerged throughout this review. Then, based on this analysis, we suggest a validation framework for industrial low-back exoskeletons. The results of this literature analysis are fully discussed in Section 4.8, also comparing the findings of similar studies. On the other hand, there are some common trends and results that are worth further discussion.

**Evaluation metrics**. EMG-based muscle activity is the metric most frequently found in the literature. Such results are in agreement among all the analyzed studies that report muscular activity reduction ranging from −10% to −40% when using the exoskeleton. On the other hand, the metrics of the functional domain showed the largest discrepancy, even when comparing results among similar tasks aided by the same exoskeleton, as discussed above (see Section 4.8). For this reason, functional metrics should be supported by metrics of other domains. A reduction of the range of motion for one joint, for example, could be interpreted as a negative effect of the device (i.e., reduced freedom of motion) while, on the other hand, allowing a torque reduction and hence less muscular effort. Subjective metrics could be used to support such objective measurements.

**Powered vs. unpowered**. As pointed out in Section 3.1, there is an increasing adoption of passive exoskeletons for industrial use, which have the advantages of being lightweight, cheaper, and easier to maintain compared to active devices. There is also an increasing interest regarding hybrid actuation, i.e., semi-active devices that combine low-power motors with elastic elements. On the other hand, sophisticated control systems are required in order to optimize the performance of these devices and make them preferable to passive designs.

**Gender unbalance**. As shown in Table 8, there is a strong gender unbalance in the test populations of the analyzed studies. Specifically, the average male-to-female ratio (M/F) is 88.89%, with 17 out of 23 studies that feature only male test subjects (100% M/F). This is likely due to the fact that no gender-specific designs have been considered in any of the analyzed exoskeletons.

**Real-world, long-term evaluation**. As mentioned above, most of the studies here analyzed reported results obtained in a laboratory setting. On the other hand, few of these highlighted how it is fundamental to consider both real-world settings and long-term usage of industrial exoskeletons. The authors of the only study performed in a real-world environment (#18) pointed out a worsening of the subjective metrics used to evaluate the exoskeleton. This was evident after a repetition of the same tasks after a four-week period, which resulted in a 25% decreased intention to use the device at work. Moreover, as discussed in #6, inter-session and intra-session reliability of the selected metrics is also an important issue to consider, especially if aiming to perform repeated evaluations over time to assess the effects of long-term use of the exoskeleton.

### 5.1. Proposed Validation Framework

In this section, we aim to suggest a validation framework for industrial low-back exoskeletons. Specifically, we propose a combination of metrics from the previously defined domains (see Section 4.2) that should allow for an exhaustive and comparative evaluation of exoskeletons and the assistance they provide to the end users. As discussed above, the lack of validation standards for industrial exoskeletons is potentially limiting their adoption in the real world. This research issue is recently starting to be tackled with joint efforts of researchers and private companies. This is the case of the EUROBENCH project (www.eurobench2020.eu), which aims to provide a consolidated benchmarking methodology for humanoid robots and lower-limb exoskeletons and prostheses [47,48]. Here, on the other hand, we identify and point out the most important metrics, starting from the analysis of the literature provided above and the reported results.

Metrics of the muscular domain should always be considered when evaluating an exoskeleton. Moreover, the muscles selected for EMG recordings should be clearly mentioned in the text to improve reproducibility. For the same reason, details about the processing of the raw EMG signals should be given in the full text. While peak and/or average RMS values are the de-facto standard, metrics that take into account the time evolution of muscular activity, such as the integral of the EMG (iEMG), should be included too. This would come for free in terms of experimental costs since it only requires additional processing of the acquired EMG signals. In addition, the choice of the muscles to consider for this analysis is important. Indeed, EMG recordings should not be limited to the muscles of the low back, which are expected to benefit from the device. Muscles of the lower limbs and the abdomen, for example, should be included in the study to investigate whether the exoskeleton induces an increase of their activity while reducing the effort on the back.

On the other hand, a comprehensive evaluation of the performance should also include metrics from other domains. In particular, here we highlight the importance of subjective metrics. Both perceived task difficulty and subjective ratings of perceived effort should be considered to gather the feedback of the end users, which is fundamental for industrial exoskeletons.

Metrics of the force/torque domain may provide insights to quantify the reduction of the stress on the lumbo-sacral joint, which is the goal for which low-back industrial exoskeletons are designed. Similar considerations can be done for metabolic metrics, as discussed above. On the other hand, metrics of these domains often require expensive or bulky instrumentation, namely force plates and gas-analysis systems, respectively, which may be difficult to use in real-world settings, outside of a laboratory.

Metrics of the functional domain should be selected and used with care. Such metrics, indeed, are often strongly device- and task-dependent. For this reason, metrics of this domain are often used to validate or assess specific design features of one exoskeleton during its performance validation. Nevertheless, some of those—kinematics, performance time, and repetition count—could also be useful for evaluating functional performance, as discussed above. Task description should be well documented in this case to improve reproducibility and better compare different results. The tasks themselves should undergo a standardization process, along with the metrics used to evaluate functional performance.

In light of what is discussed throughout this review, we end this section with a semi-qualitative chart of the five domains regarding the evaluation metrics and criteria (Figure 4). Such scores could be used to build or assess a set of selected metrics to evaluate the functional performance of industrial low-back exoskeletons.

## 6. Conclusions

Out of the 23 analyzed studies, very few comparisons among their findings could be made to evaluate the performance of low-back exoskeletons for industry. Moreover, such comparisons are hindered by differences in test population, experimental setup, assisted tasks, and many other factors. This review could not pursue a *meta-analysis* for the following reasons. First, there are too few studies that met the inclusion criteria (see Section 2). As reported above, only 2.5% of the articles found in the data collection step are considered here. Second, most of the analyzed articles illustrate pilot tests aimed at effectiveness analysis instead of more rigorous tests with higher statistical power. Third, there is a lack of homogeneity in the selected metrics to evaluate the assistive performance of industrial exoskeletons. This means that even the small percentage of studies that report statistically-robust validation results cannot be considered to compare different exoskeletons.

An industrial standard could facilitate the comparative evaluation of existing exoskeletons as well as the development of new solutions aimed to improve their weak spots. Based on a review of the state of the art, here we propose a set of metrics that should help to set up a validation framework for low-back exoskeletons, with the aim to pave the way for the definition of rigorous industrial standards. It would be interesting to apply a similar methodology to other types of exoskeletons, either for industry or for rehabilitation. Systematic reviews of upper-/lower-limb exoskeletons for industry could further accelerate the definition of evaluation standards for the benefit of the workers.

## Figures and Tables

**Figure 1 sensors-21-00808-f001:**
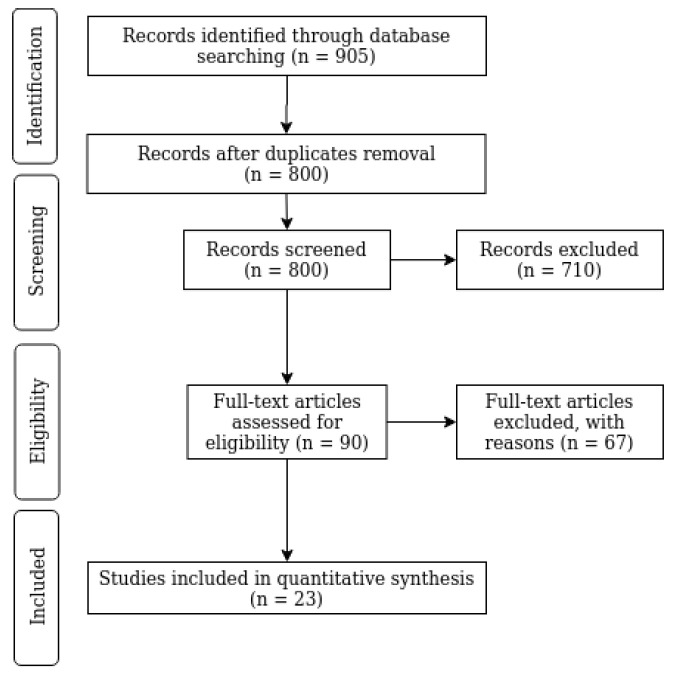
PRISMA flowchart of data collection and selection process.

**Figure 2 sensors-21-00808-f002:**
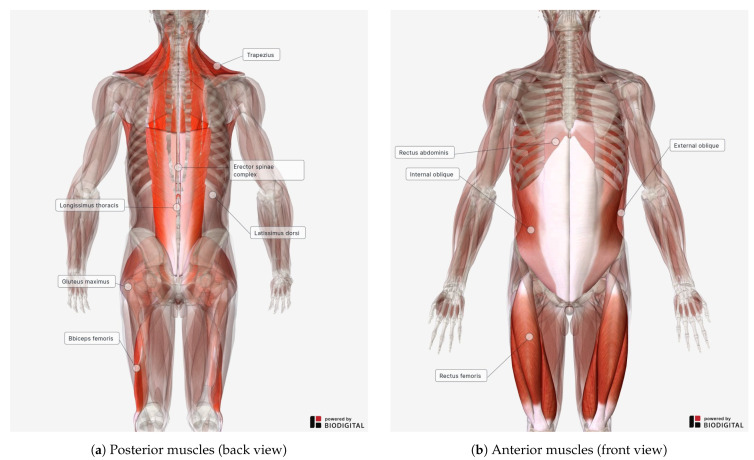
Human body anatomy highlighting (red) muscles found in the analyzed studies (see Table 7 for details). Images generated with BioDigital Human (www.biodigital.com).

**Figure 3 sensors-21-00808-f003:**
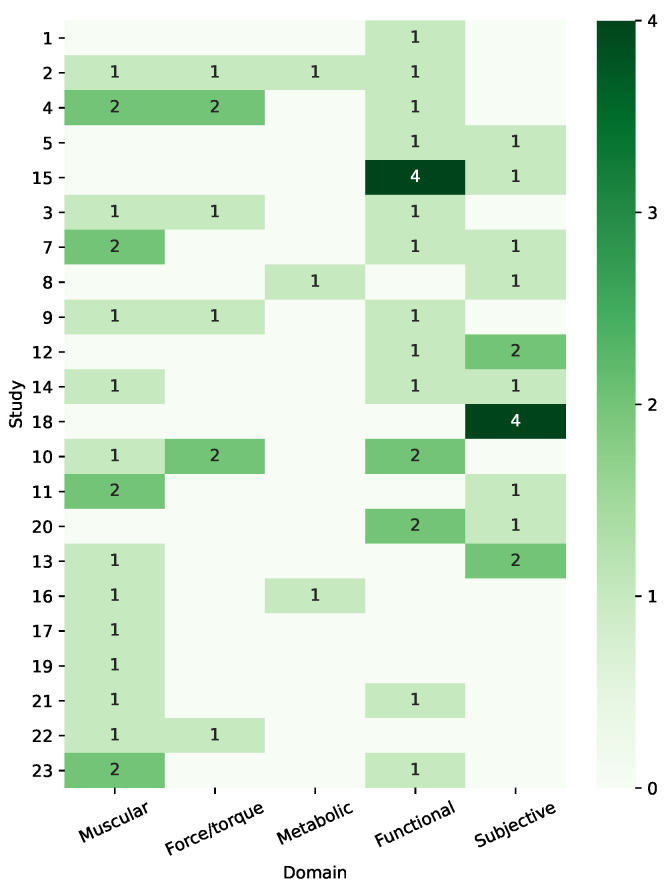
Visual representation of the sparsity of distribution of evaluation criteria and metrics among the analyzed studies, listed in the same order of Table 8. The color bar represents the number of metrics per each domain.

**Figure 4 sensors-21-00808-f004:**
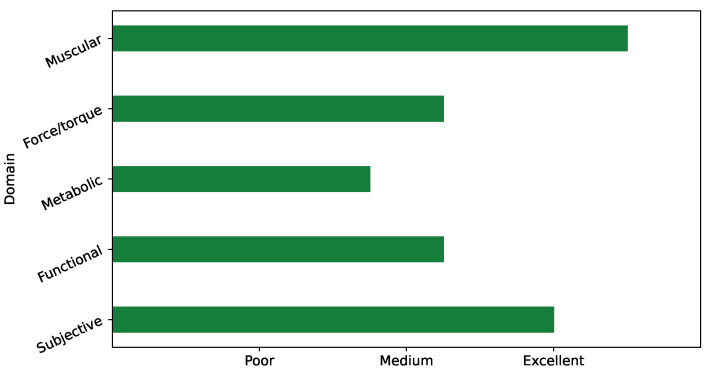
Semi-qualitative score for each of the five domains in which evaluation criteria and metrics are categorized.

**Table 1 sensors-21-00808-t001:** Keywords for database query.

Actuation	Device Type	Body Part
Active	Exoskeleton	Low-Back
Passive	Assistive device	Trunk
Hybrid	Exo-suit	Lower Limb
	Wearable device	Hip
	Wearable robot	

**Table 2 sensors-21-00808-t002:** List of applied exclusion criteria.

1.	Duplicates articles	5.	Review articles
2.	False Positive	6.	Literature not in English
3.	Exoskeletons not for low-back/trunk	7.	Exoskeletons for disabled people
4.	Studies only with simulated results		

**Table 3 sensors-21-00808-t003:** Overview of the analyzed devices. The * indicates commercial availability. *Power* indicates the maximum assistive torque [Nm]. Listed DoFs are both actuated (A-DoF), counted unilaterally (#), and non-actuated (N-DoF). Corresponding humans joint DoFs are flexion-extension (α) and adduction-abduction (β). Sensors and control data are reported only where available.

Device Name	*f*	Assisted Task(s)	Actuation	Power	A-DoF	(#)	N-DoF	Sensors	Control
**SPEXOR** * [15]	6	Static forward bending Load lifting Repetitive lifting	Passive	25 Nm	α-Hip α-L5-S1	(2)			
**Laevo** * [18]	6	Static forward bending Load lifting Load carrying Precision manual assembly	Passive		α-Hip	(1)			
**HAL** * [19]	2	Stoop load lifting Load lifting	Active	30 Nm	α-Hip	(1)		EMG Angle sensors	Hybrid EMG + Inclination (CVC + CAC)
**BackX AC** * [20]	1	Precision manual assembly	Passive		α-Hip	(1)			
**Robo-Mate** [21]	1	Load lifting	Active	40 Nm	α-Hip	(1)		EMG IMU	EMG threshold; Inclination angle Hybrid EMG + IMU
**APO** [22]	1	Load lifting	Hybrid	35 Nm	α-Hip	(1)	β-Hip	Joint encoders	α-Hip-dependent torque profile
‘**Active industrial** **exo**’ [23]	1	Load lifting Dynamic load handling	Active	40 Nm	α-Hip	(1)		Inclination sensor	Torque ∝sin(α−Hip)
**MeBot-EXO** [24]	1	Static forward bending Semi-squat lifting	Active	160 Nm	α-Hip	(1)		KinematicsInteraction force	$\dot{\alpha }$-Hip torque control w/interaction force minim.
‘**Lower-limb****exoskeleton**’ [17]	1	Squat load lifting Load carrying	Active	76 Nm	α-Hip α-Knee	(2)	α-Ankle β-Ankle	Torque sensors Joint encoders GRF sensors	(Human torque amplification (estimated by Kalman Filter)
‘**Lifting Assist** **Device (LAD)**’ [16]	1	Load lifting	Hybrid		α-Hip α-L5-S1	(2)	β-Hip β-L5-S1	IMU Potentiometers	Pre-defined force profiles (State machine)
‘**Passive spine** **exoskeleton**’ [25]	1	Dynamic bending Static forward bending	Passive	30 Nm	α-Hip	(1)			
‘**Lower extremity** **exoskeleton**’ [26]	1	Load lifting	Passive		α-Hip	(1)			

Abbreviations: DoF, Degree of Freedom; EMG, ElectroMyoGraphy; IMU, Inertial Measurement Unit; GRF, Ground Reaction Force; L5-S1, lumbo-sacral joint.

**Table 4 sensors-21-00808-t004:** Presentation of 23 analyzed studies, for which experimental setup, study type, and contribution are specified. Details about executed tasks and study population are reported in Table 8.

Study		Exoskeleton	Experimental Setup		Study Type	Contribution
			**Scenario**	**Subjects**		
1	[27]	SPEXOR	LAB	W-LBP	Pilot	Effectiveness analysis
2	[28]	SPEXOR	LL	W-MIX	Pilot	Effectiveness analysis
3	[29]	Laevo	LAB	Healthy	Pilot	Effectiveness analysis
4	[30]	SPEXOR	LAB	Workers	Pilot	Effectiveness analysis
5	[31]	SPEXOR	LAB	W-LBP	Pilot	Effectiveness analysis
6	[32]	SPEXOR	LAB	Healthy	Pilot (test–retest)	Feasibility analysis
7	[33]	BackX AC Laevo	LAB	Healthy	Pilot	Feasibility analysis
8	[34]	Laevo	LAB	Healthy	Pilot	Effectiveness analysis
9	[35]	Laevo	LAB	Healthy	Pilot	Effectiveness analysis
10	[36]	Robo-Mate	LAB	Healthy	Pilot	Effectiveness analysis
11	[37]	HAL	LL	Healthy	Pilot	Feasibility analysis
12	[38]	Laevo	LAB	Healthy	Pilot	Effectiveness analysis
13	[23]	‘Active industrial exo’	LAB	Healthy	Pilot	Effectiveness analysis
14	[18]	Laevo	LAB	Healthy	Pilot	Effectiveness analysis
15	[39]	SPEXOR	LAB	W-MIX	Pilot	Effectiveness analysis
16	[24]	MeBot-EXO	LAB	Healthy	Proof of Concept	Feasibility analysis
17	[17]	‘Lower-limb exoskeleton’	LAB	Healthy	Proof of Concept	Feasibility analysis
18	[40]	Laevo	RW	Workers	Pilot	User-acceptance analysis
19	[16]	‘Lift Assist Device’	LAB	Workers	Proof of Concept	Feasibility analysis
20	[41]	HAL	LAB	Healthy	Pilot	Feasibility analysis
21	[25]	‘Passive spine exoskeleton’	LAB	Healthy	Proof of Concept	Feasibility analysis
22	[26]	‘Lower extremity exoskeleton’	LAB	Healthy	Proof of Concept	Feasibility analysis
23	[42]	APO	LAB	Healthy	Proof of Concept	Effectiveness analysis

Abbreviations: LAB, laboratory; LL, living lab; RW, real world; W-LBP, workers with low-back pain; W-MIX, workers with and without low-back pain (mixed group).

**Table 5 sensors-21-00808-t005:** Absolute (*f*) and relative (%) frequencies of study type, contribution, scenario and test subjects for the analyzed articles presented in Table 4.

Type of Study	*f*	%
Pilot	17	73.91
Proof of Concept	6	26.09
RCT	0	0
**Contribution**	*f*	%
Effectiveness analysis	14	60.87
Feasibility analysis	8	34.78
User-acceptance analysis	1	4.35
**Scenario**	*f*	%
LAB	20	86.96
Living lab (LL)	2	8.70
Real world (RW)	1	4.35
**Test Subjects**	*f*	%
Healthy subjects (Healthy)	16	69.57
Healthy workers (Workers)	3	13.04
Workers with LBP (W-LBP)	2	8.70
Mixed workers (W-MIX)	2	8.70

**Table 6 sensors-21-00808-t006:** Absolute frequency (*f*) for each evaluation metric.

Muscular domain	*f*
Muscle activity	15
Integral of muscle activity (iEMG)	2
Average muscle activity	1
**Force/torque domain**	***f***
L5-S1 flex-ext moment	5
L5-S1 peak compression force	4
Mechanical joint work	1
Muscular force	1
**Metabolic domain**	***f***
Metabolic cost	3
**Functional domain**	***f***
Kinematics	11
Performance time	6
Posture holding time	4
Load carrying distance	3
Repetition count	3
Task performance	1
Time to extend the trunk	1
**Metabolic domain**	***f***
Metabolic cost	3

**Table 7 sensors-21-00808-t007:** Description and absolute frequency of muscles reported in analyzed studies. Muscles of the same complex are grouped together. Back (top) and front (bottom) muscles are split by a thicker line.

**Erector Spinae**	27	Longissimus Iliocostalis	9 8	Longissimus thoracis (LT) Longissimus lumborum (LL) Iliocostalis lumborum (IL)	2 1 6
**Trapezius**	1	Tr. Pars ascendens (TA)	1		
**Gluteus**	1	Gluteus maximus (GM)	1		
**Biceps femori** (BF)	3				
**Gastrocnemius**	1				
**External oblique** (EO)	7				
**Internal oblique** (IO)	3				
**Rectus abdominis** (RA)	9				
**Quadriceps femoris** (QF)	3	Rectus femoris (RF)	1		
		Vastus intermedialis (VI)	1		

**Table 8 sensors-21-00808-t008:** Evaluation metrics (domain) and reported results for the analyzed studies. Legend: ▼ significant decrease; ▲ significant increase; ▼ non-significant decrease; ▲ non-significant increase; ∼ unchanged; avg, average; ♂ male; ♀ female; 🟉 subjects with low-back pain.

#	Device	Subj.	Tasks	(Domain) Evaluation Metrics and Criteria
1	SPEXOR	19 ♂	Lifting; Repetitive bending Standing and walking Static forward bending	(Subjective) M-SFS ▼ 7%
2	SPEXOR	11 ♂	Static forward bending; Lifting Repetitive lifting; Kneeling Load carrying; Sit to stand (StS)	(Muscular) LT ▼ 10%, IL ▼ 16%, LL ▼ 16%, EO ∼, RA ∼ (Functional) Kinematics ∼ (Metabolic) Met. cost ▼ 18% (Force/torque) Joint work ▼
4	SPEXOR	10 ♂	Static forward bending Load lifting	(Muscular) IL-LL ∼, RA-EO ∼. (Force/torque) L5-S1 Fc ▼ 21%, L5-S1 Mfe ▼ 8.1% (Functional) α-Hip ▼ 17% (Muscular) ▼ 22% (avg.), (Force/torque) L5-S1 Fc ▼ 14% (Functional) α-Hip ∼
5	SPEXOR	🟉7 ♂ 🟉7 ♀	Static forward bending; Lifting Load carrying; Kneeling; Walking Sit to stand; Stair climbing	(Functional) Posture holding time: ▲12.29% for SFB; ▲8.66% for StS; ▲7.26% for climbing (Subjective) Discomfort: ▼ for SFB and sit to stand
15	SPEXOR	🟉13 ♂ 11 ♂	Static forward bending; Lifting Load carrying; Kneeling; Walking Sit to stand; Stair climbing	(Functional) Lifts/2-min *▲* 12.5%; Posture holding time ▲35.5% for SFB; Walk dist. ▼ 7.7% Perf. time ▲8.2% for stair climbing. (Subjective) PTD ▼ for SFB, lifting, kneeling
3	Laevo	11	Lifting	(Muscular) ▼ 8% (avg.) (Force/torque) L5-S1 Fc ▼ 6% (Functional) Peak $\dot{\alpha }$-Hip ▼ 17%
7	BackX AC Laevo	9 ♂ 9 ♀	Static forward bending	(Musc.) TES-IL ▼ 46%, ▼ 35% (avg.) (Func.) Perf. time *▲* 6.9% (Subjective) BORG *▲* 75% (Musc.) TES-IL ▼ 22.5%, ▼ 13% (avg.) (Func.) Perf. time *▲* 7.6% (Subjective)
8	Laevo	18 ♂	Static forward bending Walking; Sitting; Squatting	(Metabolic) For lifting: Met. cost ▼ 16.5%. (Subjective) PTD ▼ for SFB. For load carrying: Met. cost ▲ 14.5% PTD ▲ for walking, sitting, squatting
9	Laevo	11 ♂	Static forward bending	(Muscular) IL ▼, EO *▲* 10%+, IO *▲* 10%+, RA ∼, LL ∼ (Force/torque) L5-S1 Mfe ▼ (Functional) α-Hip ▼
12	Laevo	18 ♂	Static forward bending; Lifting Load carrying; Sit to stand	(Functional) Perf. time ▲ for SFB (Subjective) PTD ▼ for SFB PTD *▲* for StS and walking. LD ▼ for SFB (low back). LD ▲ for SFB (chest)
14	Laevo	9 ♂ 9 ♀	Static forward bending Static holding task (SHT)	(Muscular) For SFB: BF ▼ 20%, TA ▼ 44%, (Functional) α-Hip ▲ 15.9% ESL▼35%, ESI▼38% (Subjective) LD ▼ (low back) For SHT: BF ▼ 24%, TA ▼ 50% LD ▲ (chest) ESL▼37%, ESI▼44%
18	Laevo	30 ♂	Static forward bending Load lifting	(Subjective) LPD (over time): ▼40% (low back), ▼50% (wrist), ▲50% (chest) UMUX (over time): ∼ Donning/doffing, ▼ Task perf. Intention to use(over time) ▼ 25%
10	Robo-Mate	10 ♂	Load lifting	(Muscular) IL-LL ▼19% (Force/torque) L5-S1 Fc ▼17.8%, L5-S1 Mfe ▼ (Functional) α-Hip ▼ 15.9%, $\dot{\alpha }$-Hip ▼25%
11	HAL	14 ♂	Lifting	(Muscular) Muscle activity: TES ▼11.5%, LES▼4.5%, QF*▲*15% (Subjective) BORG ∼ iEMG: TES ▼19.55%, LES▼11.4%
20	HAL	11 ♂ 7 ♀	Stoop load lifting	(Functional) Number of lifts ▲45%; Lifting time *▲*44% (Subjective) ▼25% Perceived lumbar fatigue
13	‘Active indus-trial exo’	12 ♂	lifting; Lowering	(Muscular) RA ∼, BF▼5%, LES▼12% (w/7.5 kg) (Subjective) BORG ▼9.5% (w/7.5 kg) LES ▼15% (w/15 kg) BORG ▼11.4% (w/15 kg) SUS > 60
16	MeBot-EXO	7 ♂	Semi-squat load lifting	(Muscular) TES ▼42.5%, LES▼38.5% (Metabolic) Met. cost ▼18%
17	Lower-limb exo	5 ♂	Lifting; Load carrying	(Muscular) For lifting: VI ▼40.8%, GA▼45.3%. For carrying: VI▼45.3%, GA▼36.1%
19	LAD	1 ♂	Lifting	(Muscular) RA ▼, ES▼
21	‘Passive spine exoskeleton’	3 ♂	Dynamic bending Static forward bending	Muscular) TES ▼54%, LES▼24% (Functional) Kinematics ∼
22	‘Lower extremity exoskeleton’	5 ♂ 1 ♀	Lifting	(Muscular) ES ▼54% (Force/torque) [l]L5-S1 Fc ▼60.3% (w/4.5 kg payload) L5-S1 Fc ▼43% (w/13.6 kg payload
23	APO	5 ♂	Lifting; Lowering	(Muscular) iEMG (average iEMG): TES ▼6% (▼34.7%); LES▼15.9% (▼33%); BF▼27.4% (▲7.1%) ESI ▼3.65% (▼8.9%) RF*▲*33.7% (*▲*40.1%) (Functional) Time for trunk extension ▼19.1%

Abbreviations: Fc, compression force; LD, local discomfort; LPD, local perceived discomfort; Mfe, flexion-extension moment; M-SFS, Modified Spinal Function Sort; PTD, perceived task difficulty; SFB, static forward bending; SUS, system usability scale; UMUX, Usability Metric for User Experience. Muscles: BF, biceps femoris; ES, erector spinae; ESI, erector spinae iliocostalis; ESL, erector spinae longissimus; GA, gastrocnemius; LES, lumbar erector spinae; QF, quadriceps femoris; RA, rectus abdominis; RF, rectus femoris; TA, trapezius pars ascendens; TES, thoracic erector spinae; VI, vastus intermedialis.

## Data Availability

Not applicable.

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
