# Peer review of "Towards a Functional Performance Validation Standard for Industrial Low-Back Exoskeletons: State of the Art Review"

_sensors, 2021, doi:10.3390/s21030808_

Round 1

Reviewer 1 Report

The review is very good.  The paper is clear and the discussion and conclusions are good.

While the summary in Table 3 is good, there is more data that would help, if they are available.  What are the torque/load capabilities for each system?

What are the control modes?  It is sated EMG input, IMU, etc., but what control is used pure torque? Hybrid torque/ position? Other?

EMG is messy.  Is there information on the processing of the signals?

Author Response

Please, see the attached PDF file

Reviewer 2 Report

The work is well organized.

However, in the introduction, a deeper analysis of the available devices can be made.

I would suggest in the introduction, when describing the existing solutions for the industrial exoskeletons, to cite papers related to the different available solutions: lower limbs, upper limbs, and back support. Some example citations can be found below:

LLE:

[1] Luger, Tessy, et al. "Subjective evaluation of a passive lower-limb industrial exoskeleton used during simulated assembly." IISE Transactions on Occupational Ergonomics and Human Factors 7.3-4 (2019): 175-184.

[2] Sankai, Yoshiyuki. "HAL: Hybrid assistive limb based on cybernics." Robotics research. Springer, Berlin, Heidelberg, 2010. 25-34.

ULE:

[3] Mauri, Alessandro, et al. "Mechanical and control design of an industrial exoskeleton for advanced human empowering in heavy parts manipulation tasks." Robotics 8.3 (2019): 65.

[4] Pacifico, Ilaria, et al. "An Experimental Evaluation of the Proto-MATE: A Novel Ergonomic Upper-Limb Exoskeleton to Reduce Workers' Physical Strain." IEEE Robotics & Automation Magazine 27.1 (2020): 54-65.

BSE:

[5] Roveda, Loris, et al. "Design methodology of an active back-support exoskeleton with adaptable backbone-based kinematics." International Journal of Industrial Ergonomics 79 (2020): 102991.

[6] Yang, Xiaolong, et al. "Spine-inspired continuum soft exoskeleton for stoop lifting assistance." IEEE Robotics and Automation Letters 4.4 (2019): 4547-4554.

In addition, for lower limbs exoskeleton validation standard, consider citing the EUROBENCH project, developing a facility for the test and benchmark of LLE: http://eurobench2020.eu/

W.r.t. EUROBENCH, the following papers can be found describing both testbeds and metrics for performance evaluation:

[7] Nicole Maugliani et al. "Lower-Limbs Exoskeletons Benchmark Exploiting a Stairs-Based Testbed: the STEPbySTEP Project." Werob 2020.

[8] Torricelli, D., and Jose L. Pons. "EUROBENCH: Preparing robots for the real world." International Symposium on Wearable Robotics. Springer, Cham, 2018.

[9] Rainieri, Giuseppe, et al. "Usability and Interfaces of Lower Limb Exoskeletons: a Framework for Assessment and Benchmark."

[10] Taborri, Juri, et al. "BEAT: Balance Evaluation Automated Testbed for the standardization of balance assessment in human wearing exoskeleton." 2020 IEEE International Workshop on Metrology for Industry 4.0 & IoT. IEEE, 2020.

[11] Pasinetti, Simone, et al. "Validation of Marker-Less System for the Assessment of Upper Joints Reaction Forces in Exoskeleton Users." Sensors 20.14 (2020): 3899.

Even if the paper is related to the back exoskeleton validation standards, consider comparing the state of the art procedure for such devices with the other devices validation procedures to highlight the main differences and common approaches.

Author Response

Please, see the attached PDF file

Reviewer 3 Report

This manuscript provides a review of the state of the art of the performance evaluation of the low-back exoskeletons. It reports the criteria used by the authors to assess the quality of the exoskeletons. The different metrics are presented demonstrating the lack of gold standard in the domain of exoskeletons design and evaluation. The paper focuses on the one hand on a presentation of the methodology used by the researchers to evaluate the exoskeletons and, on the other hand, on proposition for standard methods for the evaluation of the exoskeletons. In detail, 23 peer-reviewed articles focusing only on different low back exoskeletons were analyzed in this review. The manuscript is very interesting and clear for the researchers in the field. However, the manuscript contains oversights and shortcomings in the different parts that should be addressed to strengthen the quality of the reported research. The reviewer has outlined these below in the comment sections. All pages and lines refer to pages and lines in the body of the manuscript, unless otherwise noted. Please address this in your revised manuscript.

Comments

It is unclear for the reviewer why the authors focus on low back exoskeletons, ignoring the exoskeletons for the other parts of the body or for disabled people. However, the technologies are similar and the metrics used for such exoskeletons are equivalent to those described in this manuscript. It could enlarge the database and increase the number of available papers.

For the reviewer, it is very unclear why passive exoskeletons decrease metabolic costs and muscle activity. Indeed, such exoskeletons provide no power at the musculoskeletal system. Thus, for a given task (i.e. lifting), the necessary work is equivalent and the requested power is always the same while it can be redistributed differently between the joints. Accordingly, if some muscles activity decrease, probably other muscles have an increasing activity. Thus, in the proposed validation framework, the number of analyzed muscles should be extended. In the same way, the flexion-extension moment mainly decreases in the presented research. However, the authors must present the torques at the others joints. Indeed, a decrease moment at L5-S1 level could lead to an increase moment at the knee level for example. Finally, using the usual inverse dynamics process to compute joint torques, the researchers have to merge kinematics and moments to compute joint power at the different levels and compare the subject behavior with and without the exoskeletons. To conclude this comment, the reviewer thinks that, particularly for passive exoskeletons, the authors of this manuscript must recommend the dynamics study of the body at a larger scale than only the concerned joint.

Minor comments

Chapter 3

Page 6 - Table 3: In the column “Assisted tasks”, the authors mention “dynamic lifting”. Is the term “dynamic” appropriate? Aren't lifts inherently dynamic?

Chapter 4

Page 10 – l281-290: No domain concerns the kinematics characterization of the exoskeletons neither, joint power developed by the subject and by the apparatus. However, page 11, line 334, the authors report a functional domain (not cited previously) with kinematics evaluation. Perhaps, the authors should be clearer. Why the functional domain is not cited in the introduction of the domains on page 10?

Author Response

Please, see the attached PDF file

Reviewer 4 Report

The paper addresses a very interesting topic, that of exoskeleton systems for back support. The authors note the lack of standardization and regulation to facilitate the adoption of these systems in the industrial environment. There is a large volume of work done on the manuscript. The paper is well structured, with recent references. I consider that the paper brings a consistent scientific contribution and I recommend its publication.

Author Response

Please, see the attached PDF file

Round 2

Reviewer 2 Report

The paper can now be accepted.